# Panorama Generation with Multiple Custom Background Objects

## Abstract

While text-to-image and customized generation methods demonstrate strong capabilities in single-image generation, they fall short in supporting immersive applications that require coherent 360° panoramas. Conversely, existing panorama generation models lack customization capabilities. In panoramic scenes, reference objects often appear as minor background elements and may be multiple in number, while reference images across different views exhibit weak correlations. To address these challenges, we propose the first diffusion-based framework for customized multi-view image generation. Our approach introduces a decoupled feature injection mechanism within a dual-UNet architecture to handle weakly correlated reference images, effectively integrating spatial information by concurrently feeding both reference images and noise into the denoising branch. A hybrid attention mechanism enables deep fusion of reference features and multi-view representations. Furthermore, a data augmentation strategy facilitates viewpoint-adaptive pose adjustments, and panoramic coordinates are employed to guide multi-view attention. Experimental results demonstrate our model's effectiveness in generating coherent, high-quality customized multi-view images.

## 1 Introduction

In recent years, text-to-image generation models Rombach et al. (2022); Podell et al. (2023); Betker et al. (2023); Mokady et al. (2023); Saharia et al. (2022) have made significant breakthroughs, greatly advancing the field of image generation. Subsequently, customized image generation methods Karras et al. (2019); Song et al. (2024); Ma et al. (2024); Zhang et al. (2025); He et al. (2025) such as DreamBooth Ruiz et al. (2023) and Custom Diffusion Kumari et al. (2023) have emerged, allowing users to generate images based on specific references, aiming to meet customized requirements while preserving key features from the references. However, these studies primarily focus on single-image generation. In contrast, immersive applications such as virtual reality Yang et al. (2022; 2024) and digital twins rely on coherent spatial perception, which often requires generating continuous multi-view images to construct a 360° panoramic environment. Methods like MVDiffusion Tang et al. (2023) and Panfusion Zhang et al. (2024) can generate panoramas Wu et al. (2023); Li et al. (2024b); Chen et al. (2022); Ni et al. (2025); Gao et al. (2024) or multi-view images from text, but they cannot incorporate user-specified images for customization. Although MV-Adapter Huang et al. (2024a) can generate multi-view images of a given object, it is essentially an object-level multi-view reconstruction model Tang et al. (2024); Huang et al. (2024b); Liu et al. (2023); Long et al. (2024) and struggles to integrate objects naturally into complex panoramic scenes.

Furthermore, a common characteristic of both single-image customization and multi-view object customization is that the reference object occupies the dominant part of the image, and only one reference object can be generated, with little or no background interference. And in object-level multi-view generation, the correlations between images from different views are very high, allowing them to share features from the same reference image. In contrast, panoramas typically depict a scene where there is no dominant object; almost all elements appear as background components, and different views may contain different reference objects. This results in weaker correlations between images. Additionally, if a reference object appears in overlapping views, consistency across different perspectives must be maintained. These factors collectively increase the difficulty of customized panorama generation.

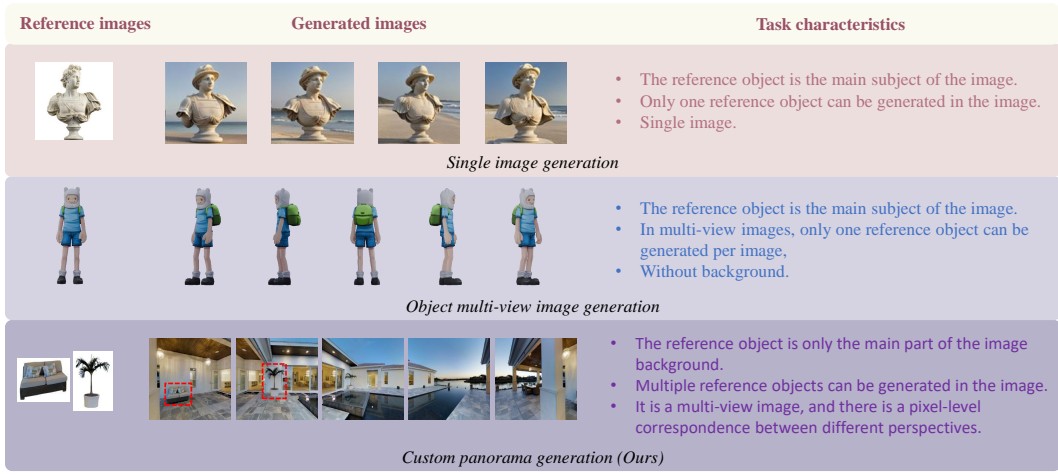

Figure 1: Differences between various tasks.

Moreover, datasets for customized panorama generation are relatively scarce. First, compared to single images, panoramic image datasets for scene generation are inherently limited. Second, training customized generation models usually requires multiple paired images, which is even more challenging to obtain for panoramic data.

This paper focuses on customized panorama generation. We define the task as follows: Given one or more reference images and a textual description, the model must generate a sequence of continuous multi-view images with pixel-level correspondence. These images should be seamlessly stitched together to form a complete 360° panoramic view. Simultaneously, the target object from the reference images should be naturally and consistently integrated into the scene from specified viewpoints while retaining its appearance characteristics. We provide a visual comparison of our setting with existing tasks in Figure 1.

To address this task, we propose an innovative solution. At its core, since the reference object in a panorama may occupy only a local region and undergo significant viewpoint changes, leading to weak inter-view correlations, we design a decoupled feature injection mechanism within a dual-UNet architecture to separately handle scene generation and object feature preservation. We introduce a hybrid attention mechanism to achieve deep fusion of reference features and multi-view representations. Additionally, a data augmentation strategy is employed to adapt the model to various object poses. Furthermore, panoramic coordinates are utilized to explicitly guide attention across multiple views. Experimental results demonstrate that our model effectively generates high-quality, coherent customized multi-view images.

## 2 METHOD

### 2.1 PIPELINE

The input of our model includes reference images, noise, panoramic coordinates, and multimodal prompts, where multimodal prompts are achieved by embedding the CLIP Radford et al. (2021) features of the images into the text. The output of the model is continuous images from different viewpoints. The workflow begins by providing one or more reference objects and their positions in the images. These reference objects are then pasted into the corresponding positions to generate reference images that come from different viewpoints or different objects. Next, these reference images are encoded using a Variational Autoencoder Kingma & Welling (2022). The encoded images are then fed into a dual UNet architecture with a decoupled feature injection mechanism, which consists of a denoising branch and a reference branch. The denoising branch takes the encoded reference images concatenated with noise as input to enhance image quality and provide spatial guidance. The reference branch provides reference image features. The two branches process these features

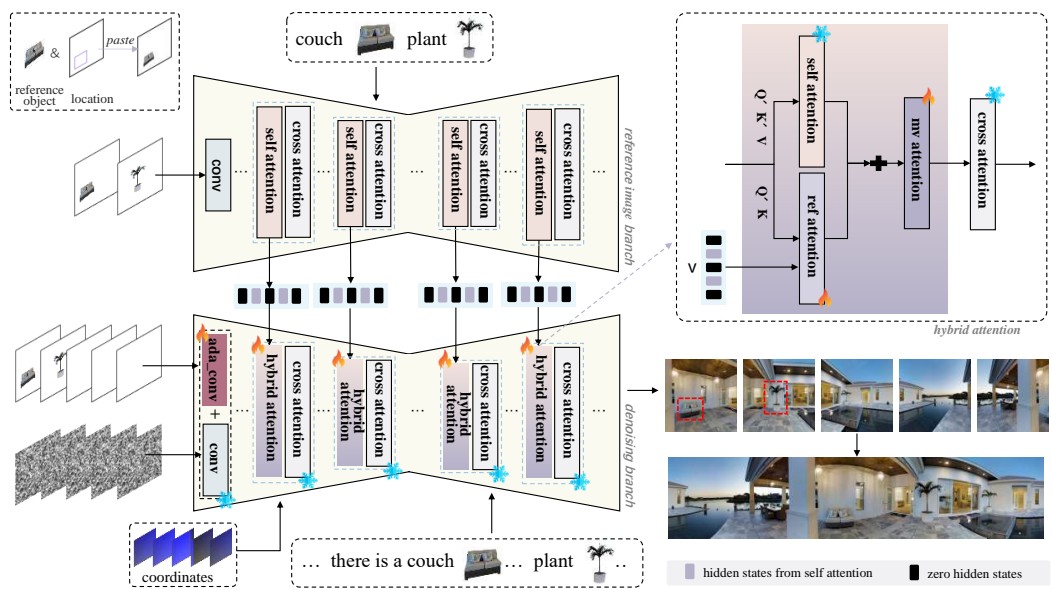

Figure 2: Pipeline. Reference images are fed into a dual-UNet architecture comprising denoising and reference branches. The denoising branch concatenates encoded images with noise to enhance quality and provide spatial guidance, while the reference branch extracts image features. Features from both branches are fused via a hybrid attention mechanism, with panoramic coordinate guidance ensuring viewpoint consistency.

in parallel. Features extracted from both branches are integrated using a hybrid attention mechanism. Panoramic coordinate guidance and row-wise self-attention Li et al. (2024a) is used to ensure viewpoint consistency.

The term "reference image" denotes the original target-only image in multimodal prompts, and otherwise represents a 512×512 composite image with the resized object positioned as specified, with this unified definition applied consistently throughout the paper.

## 2.2 DECOUPLED FEATURE INJECTION

In panoramic image generation , reference images from different viewpoints typically contain distinct objects with no inherent correlation. Therefore, unlike multi-view object generation, the same reference images cannot be used across different viewpoints. To tackle this issue, a decoupled feature injection mechanism is introduced within the dual-UNet architecture, aimed at managing weakly correlated reference images from different viewpoints, as shown in Figure 2. This mechanism processes reference images adaptively: when a reference is available for a target viewpoint, its features are extracted and injected into the denoising branch. Otherwise, zero-initialized feature vectors are employed to maintain network integrity. This conditional approach allows flexible adaptation to references at arbitrary viewpoints, while ensuring stable generation in their absence. The feature injection is formally defined as:

$$\mathbf{f}_v = \begin{cases} \mathcal{E}(I_v), & \text{if reference } I_v \text{ exists} \\ \mathbf{0}, & \text{otherwise} \end{cases} \tag{1}$$

where $\mathcal{E}$ denotes the feature extraction module. $f_v$ denotes the injected features for viewpoint $v$.

## 2.3 HYBRID ATTENTION

MV-Adapter Huang et al. (2024a) employs a parallel structure to the original self-attention layer to fuse reference image features, with both its keys and values derived from reference image features

and queries coming from noise branch features:

$$f_{\text{self}} = \text{SelfAttn}(f_{\text{in}}) + \text{MVAttn}(f_{\text{in}}) + \text{RefAttn}(Q = f_{\text{in}}, (K, V) = f_{\text{ref}}) + f_{\text{in}} \qquad (2)$$

where $f_{\text{in}}$ is the input to the attention block, $f_{\text{ref}}$ is the reference image features, SelfAttn is the original self-attention layer from the T2I model, MVAttn is the multi-view attention layer, and RefAttn is the reference image cross-attention layer, both created by duplicating the spatial self-attention layer.

our model operates on multi-view images without explicitly injecting reference image features into every viewpoint. The parallel architecture of MV-Adapter suffers from progressive feature degradation, where visual cues present in the current view tend to deteriorate or disappear in adjacent generated views. To address this, we employ a hybrid attention mechanism that first performs deep fusion of reference image features and then enforces cross-view consistency constraints. Furthermore, our denoising branch also takes reference image as input. The denoising branch is capable of computing attention scores by calculating similarities within itself, capturing the intensity of the reference image features that need to be obtained. It then retrieves these features from the reference image branch. Therefore, our keys and queries are derived from the noise branch features, and the values are derived from the reference image features. The aforementioned procedure can be mathematically formulated as:

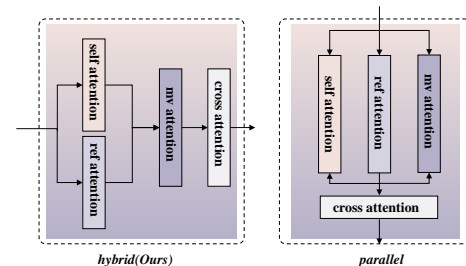

Figure 3: hybrid(ours) vs parallel.

$$f_{\text{in}} = \text{SelfAttn}(f_{\text{in}}) + \text{RefAttn}((Q, K) = f_{\text{in}}, V = f_{\text{ref}}) \qquad (3)$$

$$f_{\text{self}} = \text{MVAttn}(f_{\text{in}}) + f_{\text{in}} \qquad (4)$$

## 2.4 MULTIMODAL PROMPT INPUT

Our model incorporates reference images as input and employs a text-image multi-modal prompting mechanism that enables effective collaboration between prompt features and reference image features to facilitate target object generation. Textual information provides semantic guidance, while reference images supply visual cues. For the reference image branch, we use multi-modal prompts when reference images are provided, otherwise, an empty string. We encode images using CLIP and introduce a linear layer on image embeddings to facilitate modality alignment.

## 2.5 DATA AUGMENTATION

To enable our model to place objects at specified locations and adaptively adjust their pose, we faced the challenge that typical panoramic datasets lack images where only the object's pose changes. To address this, we employed a data augmentation strategy on reference images, applying techniques such as random cropping, minor rotations ($\pm 15°$), horizontal flipping, and geometric deformations (perspective and affine transformations with a maximum distortion of $5\%$).

## 2.6 PANORAMA COORDINATE GUIDANCE

MV-Adapter relies on camera-to-object distance parameters, making it unsuitable for panorama generation due to significant depth variation among objects. Instead, we adopt equirectangular panoramic coordinates for implicit positional encoding. By computing the spherical coordinates of panorama pixels, we achieve a consistent scene representation without requiring explicit distance information. For a panoramic image with resolution $W \times H$, the spherical coordinates $(\theta, \phi)$ of each pixel $(u, v)$ are computed as follows:

$$\theta(u) = \frac{2\pi u}{W} \qquad (5)$$

$$\phi(v) = \frac{\pi v}{H} - \frac{\pi}{2} \qquad (6)$$

Here, $\theta \in [0, 2\pi]$ denotes the azimuth angle, and $\phi \in [-\pi/2, \pi/2]$ represents the elevation angle. This mapping ensures continuity at the panorama boundaries by aligning $\theta = 0$ with $\theta = 2\pi$. To store the spherical coordinates as images, we normalize them into the $[0, 255]$ range:

$$\theta_{\text{norm}} = \frac{\theta}{2\pi} \times 255 \tag{7}$$

$$\phi_{\text{norm}} = \frac{\phi + \pi/2}{\pi} \times 255 \tag{8}$$

The normalized coordinate panorama is then converted into five perspective views, serving as implicit positional guidance for consistent multi-view generation.

## 3 EXPERIMENTS

### 3.1 EXPERIMENTAL SETUP

We conduct experiments on the Matterport3D Chang et al. (2017) dataset, which contains 10,912 indoor panoramas. Following the MVDiffusion data split, 9,820 panoramas are used for training and 1,092 for evaluation. Each panorama is projected with a FoV of 90° and a yaw step of 72°. Experiments are conducted with 5 and 15 viewpoints. In the 5-view setting, we use the five central horizontal viewpoints. For the 15-view setting, additional rows are added above and below the central horizontal viewpoints, resulting in three vertical rows with five horizontal viewpoints per row. Captions are generated using BLIP-2 Li et al. (2023), and target objects are detected using YOLOv8n Jocher et al. (2023), with only high-confidence and size-qualified objects from selected categories retained. The 5-view model is trained on four NVIDIA RTX 3090 GPUs (24GB VRAM), and the 15-view model on four NVIDIA RTX 4090 GPUs (48GB VRAM).

We employ Fréchet Inception Distance (FID) Heusel et al. (2017), Inception Score (IS) Salimans et al. (2016), CLIP Score (CS) Radford et al. (2021), and Peak Signal-to-Noise Ratio (PSNR) as our evaluation metrics.

### 3.2 CUSTOMIZED MULTI-VIEW IMAGE GENERATION

**Baseline.** Our model has multi-view image generation capability, and we selected three related methods for comparison. A brief description is as follows:

*Text2Light* Chen et al. (2022) is a zero-shot model for generating panoramas from text.

*MVDiffusion* Tang et al. (2023) is a multi-view image generation model, but it generates eight views (FOV=90°, ROT=45°).

*PanFusion* Zhang et al. (2024) is an end-to-end model trained on Matterport3D Chang et al. (2017) to generate panoramic images.

We conduct experiments under 5-view and 15-view settings, generating the corresponding perspective views from each model's output for comparison. Column-wise attention is incorporated in the 15-view model to enforce vertical viewpoint consistency.

Table 1: Comparison with SoTA methods in 5-view setting.

| Method | Custom | Generation Quality | | | | Inference Efficiency | |
|---|---|---|---|---|---|---|---|
| | | FID↓ | IS↑ | CS↑ | PSNR↑ | GPU Mem.(GB)↓ | Time(s)↓ |
| Text2Light | ✗ | 54.67 | 5.53 | 23.97 | 8.05 | **3.9** | 65 |
| MVDiffusion | ✗ | – | – | – | – | 7.9 | 46 |
| PanFusion | ✗ | 25.39 | 6.64 | 25.08 | 10.91 | 15.4 | 46 |
| Ours (w/ Ref.) | ✓ | **20.91** | **6.70** | **25.79** | **11.40** | 6.5 | **10** |
| Ours (w/o Ref.) | ✓ | **21.79** | **6.69** | **25.54** | **11.40** | 6.5 | **10** |

Prompt: doorway leading to a hallway with a painting on the wall, there is a large living room with a couch and a large window, arafed view of a living room with a couch, chair, and a potted plant 🪴, there is a living room with a couch, chair, table and a fireplace, there is a picture of a living room with a fireplace and a couch

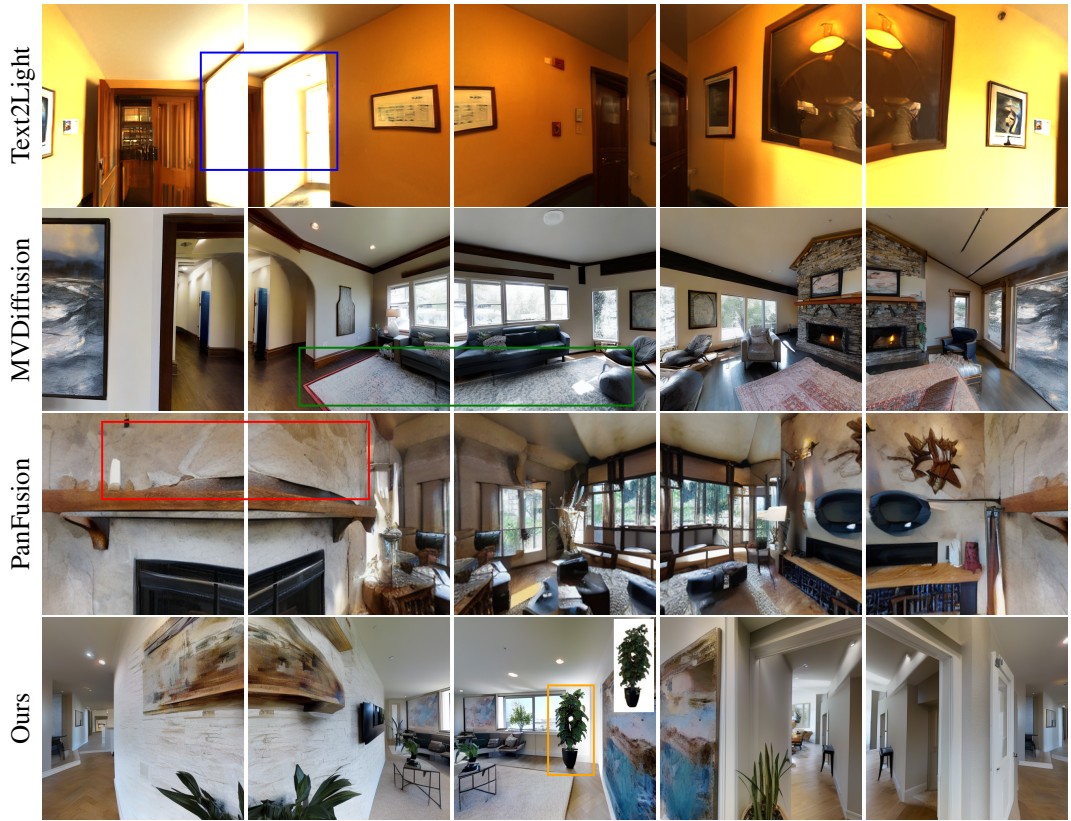

Figure 4: 5-view generation results of different methods, with problematic regions in baseline methods highlighted using colored boxes: inconsistent lighting, color mismatch, incomplete or blurry objects. Our method mitigates these issues and provides a customized multi-view image.

**Results.** Table 1 demonstrates that our model trained with 5 views surpasses all baseline methods across FID, IS, CS, and PSNR metrics. And our model achieves the fastest inference speed with low GPU memory consumption. Table 2 compares the model trained with 15 views under 15-, 5-, and 8-view configurations, showing superior or competitive performance across all configurations. Furthermore, we report results for two reference-free models, which outperform baseline methods in the 5-view setting and demonstrate strong performance in the 15-view configuration. Although our model was not trained on 8 views, it still obtains the best FID and CS, indicating strong generalization ability. A vertical comparison under the 5-view evaluation setting shows that the model trained on 15 views slightly underperforms the one trained specifically on 5 views. This may be due to the non-central views in the Matterport3D dataset containing more blurry or structurally simple content, which leads the model to learn less informative patterns.

Figure 4 and Figure 5 present qualitative comparisons of the 5-view and 15-view models, along with results from various baseline methods. In particular, the results of the 15-view model are presented using both full panoramas and zoomed-in perspective views from specific locations. Compared to Text2Light, our method produces more natural lighting and richer scene content. Compared to MVDiffusion, our model ensures better continuity of the same object across different viewpoints. Compared to PanFusion, our results exhibit higher realism and more complete object structures. Additionally, unlike previous methods that rely on a single reference image, our model is capable of incorporating multiple reference images, enhancing its flexibility and ability to generate more diverse customized views.

Prompt: a view of a large room with a potted plant and a window, arafed living room with a chandelier and a couch, arafed living room with a couch 🖐, chairs, a table, and a chandelier, there is a plant 🌳 in a pot in a room with a window, there is a view of a courtyard through a window of a house

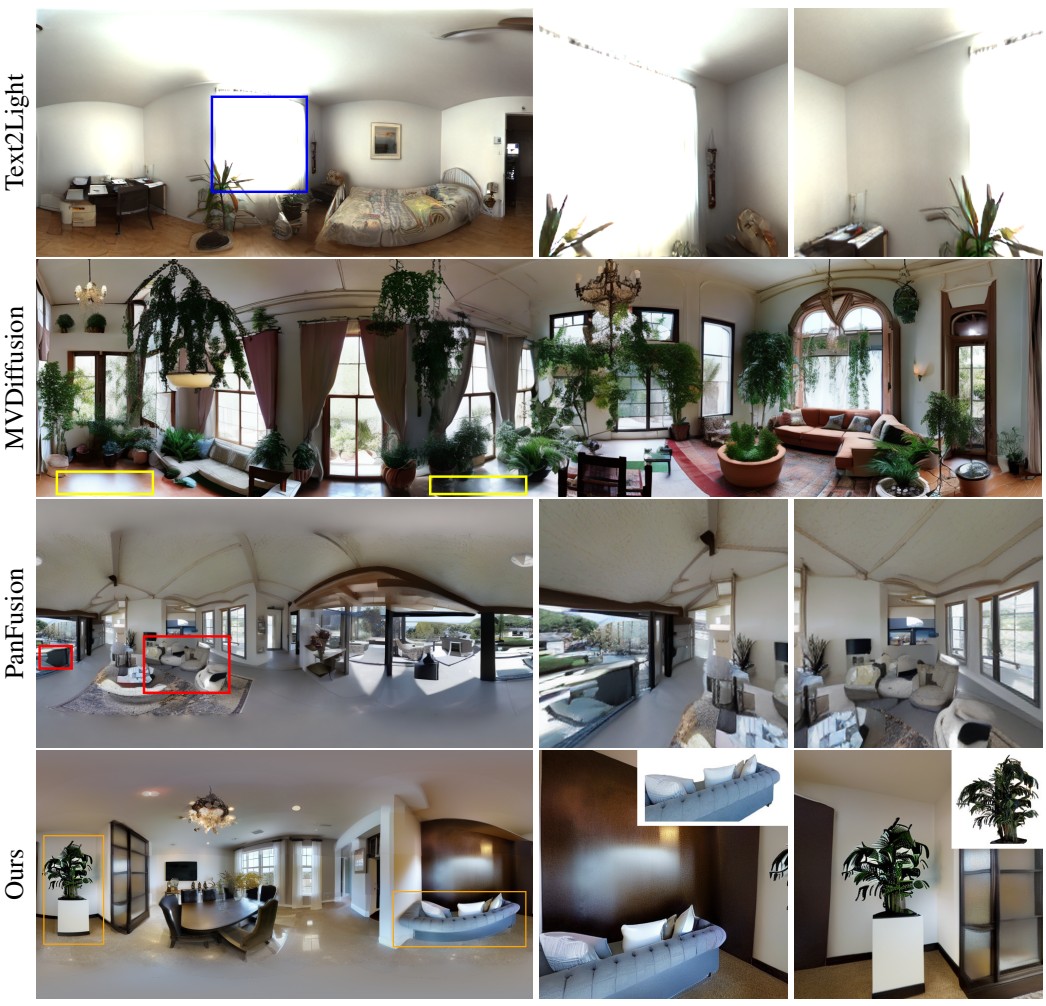

Figure 5: 15-view generation results of different methods. The generated panorama highlights problematic regions in baseline methods with colored boxes: inconsistent lighting, color mismatch, and incomplete or blurry objects. Our method mitigates these issues and provides a customized multi-view image. Perspective projections for some issues are provided.

Table 2: Comparison with SoTA methods in 15-view setting.

| Method | 15 views | | | 5 views | | | 8 views | | |
|--------|----------|----------|----------|----------|----------|----------|----------|----------|----------|
| | FID ↓ | IS ↑ | CS ↑ | FID ↓ | IS ↑ | CS ↑ | FID ↓ | IS ↑ | CS ↑ |
| Text2Light | 53.50 | 6.45 | 23.78 | 54.75 | 5.48 | 23.93 | 51.84 | 5.73 | 23.76 |
| MVDiffusion | - | - | - | - | - | - | 25.27 | **6.90** | 26.34 |
| PanFusion | 24.84 | **7.03** | 24.64 | 26.21 | 6.63 | 24.21 | 23.84 | 6.72 | 24.22 |
| Ours(w/ Ref.) | **20.93** | 6.92 | **27.51** | **25.65** | 6.76 | **28.01** | **23.55** | 6.16 | **26.80** |
| Ours(w/o Ref.) | **21.50** | 6.99 | **27.53** | 26.72 | **6.99** | **27.99** | 24.51 | 6.50 | **26.72** |

### 3.3 ABLATION STUDY

Ablation experiments are conducted under the 5-view setting. Specifically, we compare two variants: one without the data augmentation mechanism, and another where the hybrid attention structure is replaced with a parallel attention design, evaluating the contributions of each component.

*w/o Aug.* In this experimental configuration, the reference image is simply resized and placed at a fixed location without any data augmentation. All other settings remain unchanged.

*Parallel Attn.* In this experimental configuration, only the hybrid attention is replaced with parallel attention, with data augmentation retained. All other settings are kept the same.

Prompt: there is a car 🚗 that is parked in the driveway of a house, there is a hallway with a door and a planter on the floor, there is a door and a car 🚗 and a hand rail, there is a wooden wall with a metal pole on it, there is a car parked in a driveway next to a building

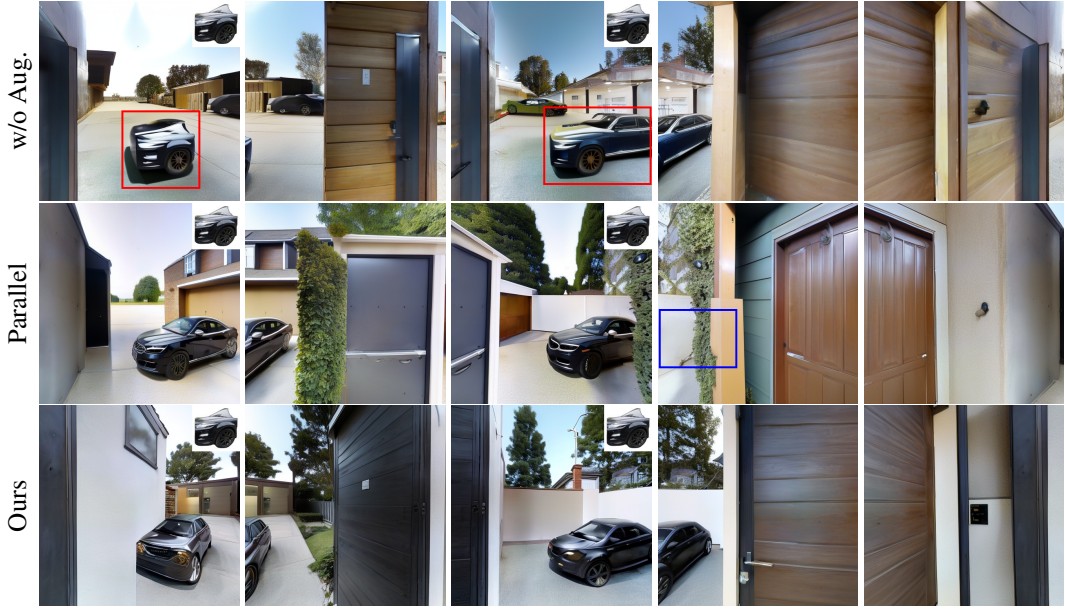

Figure 6: Ablation study results. We show the perspective views of 5 consecutive viewpoints from different settings, with problematic regions in baseline methods highlighted using colored boxes: incomplete or failed view-adaptive, feature degradation.

**Results.** Table 3 presents the results of the ablation study, demonstrating that the proposed components improve the overall performance. The visualization results in Figure 6 show that our hybrid attention module effectively merges reference image features with multi-view consistency constraints, ensuring continuous viewpoint alignment. Additionally, our data augmentation mechanism equips the model with viewpoint-adaptive capabilities.

Table 3: Ablation study of model components under the 5-view setting.

| Method | FID↓ | IS↑ | CS↑ | PSNR↑ |
|---|---|---|---|---|
| w/o Aug. | 23.63 | 5.94 | 25.59 | 11.09 |
| Parallel Attn. | 21.47 | 6.34 | **25.90** | 10.82 |
| Ours(w/ Ref.) | **20.91** | **6.70** | 25.79 | **11.40** |

*Hybrid attention.* Compared to the parallel architecture, our hybrid attention achieves better FID, IS, and PSNR. As shown in Figure 6, parallel architecture suffers from feature degradation during viewpoint transitions when reference images appear in overlapping view regions. This issue stems

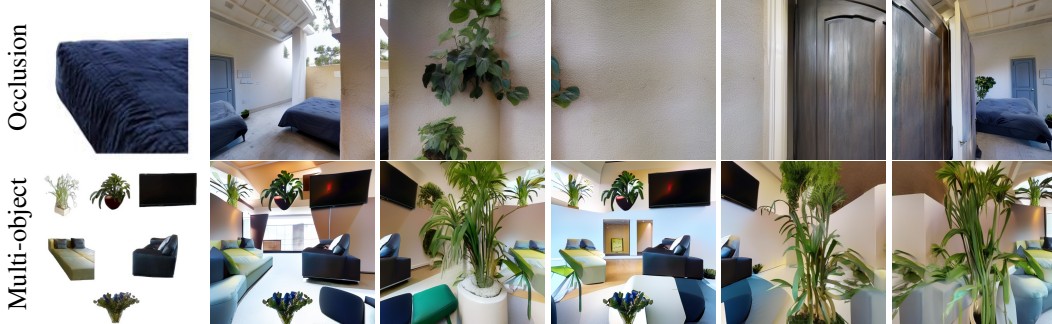

Figure 8: Visualization of some edge cases. The first row shows beds with generated occlusion, and the second row presents multi-object multi-view scenarios.

from the parallel attention architecture's decoupled mechanism for reference feature injection and multi-view consistency constraints. Essentially, while the model enforces view consistency, it does not explicitly apply these constraints to the reference image features. As a result, the reference image features needed for adjacent views are obtained implicitly through the model's multi-view consistency mechanisms, lacking persistent cross-view guidance. This leads to the degradation of key visual cues from the reference image during view transitions. In contrast, our hybrid attention mechanism first fuses the reference image features with the original features before applying multi-view constraints. This explicitly propagates the visual information from the reference image to other views, ensuring the persistent injection of reference features. As a result, it preserves detailed features from the reference image while enhancing inter-view coherence, leading to smoother viewpoint transitions and higher-quality image generation.

*Data augmentation.* Quantitative results demonstrate that the data-augmented model achieves comprehensive performance improvements across all four evaluation metrics compared to its non-augmented counterpart. Visualization results in Figure 6 show that the model without data augmentation simply copies the reference image and fails to adapt to viewpoint changes. In contrast, the model with data augmentation effectively adjusts object morphology to respond to perspective variations and can faithfully reconstruct complete objects from incomplete references.

### 3.4 FURTHER ANALYSIS

Our model has the ability to handle occlusion. In Figure 8, given a partially occluded bed with the prompt "behind a wall", the model accurately reconstructs the occluded scene, and can also generate a complete bed from only a partial input and a basic object prompt. The model also supports multi-object placement from diverse viewpoints, even in unconventional layouts, by generating plausible support structures. It is also effective at handling cases involving tiny reference objects as shown in Figure 7.

We believe that allowing users to decide the position of reference objects offers greater flexibility compared to letting the model determine object positions autonomously, as the latter often leads to unpredictability. However, considering various application scenarios, we also provide a model that autonomously determines object positions. Specific modifications and experimental results can be found in appendix B.6.

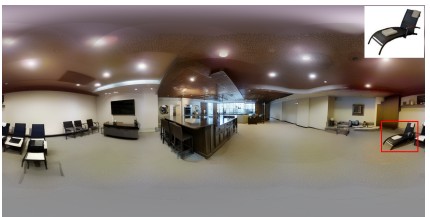

Figure 7: Tiny object case.

## 4 REPRODUCIBILITY STATEMENT

Anonymized access to our code implementation, along with comprehensive experimental details, is provided in the appendix.

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

# A RALATED WORK

## A.1 PANORAMA GENERATION

The work on multi-view scene generation can be primarily categorized into two approaches: 2D-based methods and 3D-based methods. In the former, 2D images are directly generated by considering the 3D geometric constraints between different viewpoints. MVDiffusion Tang et al. (2023) uses 8 parallel branches to generate images from 8 different viewpoints simultaneously and employs a Correspondence-Aware Attention (CAA) block to ensure viewpoint consistency. However, it requires predefined camera poses. CamFreeDiff Yuan et al. (2024) estimates the camera pose by predicting the homography transformation from the input view to a predefined canonical view, removing the need for predefined camera poses. PanoDiff Wang et al. (2023) generates 360° panoramas from unregistered narrow field-of-view (NFoV) inputs. It estimates overlap between input pairs, projects them onto a panoramic canvas, and uses a latent diffusion model (LDM) to complete the panorama. AOGNet Lu et al. (2024) introduces an autoregressive framework for 360° image generation using progressive outpainting with NFoV images and text guidance, but it lacks fine-grained control. PanFusion Zhang et al. (2024) uses two parallel branches to process panoramic and perspective images separately, and employs the Equirectangular Projection Perception (EPP) module to enable the model to capture correspondences between panoramic and perspective views. 3D-based methods reconstruct the scene in 3D space and generate multi-view images through rendering. Sat2Scene Li et al. (2024b) integrates point clouds with a 3D diffusion model in three steps: extracting the scene's geometry, applying sparse convolutions to color the foreground point cloud, generating the background with a 2D model, and using neural rendering to create high-quality images from arbitrary viewpoints.

## A.2 CUSTOMIZED IMAGE GENERATION

Currently, research on customized generation mainly focuses on two levels: single-image customization Ruiz et al. (2023); Ye et al. (2023); Yang et al. (2023); Chen et al. (2024) and object-level multi-view Huang et al. (2024a) customization. These methods either generate customized images without maintaining multi-view consistency, or produce view-consistent images without customization. DreamBooth Ruiz et al. (2023) customizes text-to-image diffusion models by fine-tuning them on a few images of a subject. It learns to associate a unique identifier with the subject, enabling the generation of realistic and diverse images of that subject in new contexts. IP-Adapter Ye et al. (2023) introduces image prompting through a decoupled cross-attention Vaswani et al. (2017) mechanism that independently attends to text and image features, allowing more flexible integration of visual prompts. Paint by Example Yang et al. (2023) incorporates a reference image by encoding it with CLIP Radford et al. (2021) and injecting the features into the diffusion model using cross-attention, with spatial control guided by a mask. AnyDoor Chen et al. (2024) enables object placement at specified locations and shapes within an image. It uses high-pass filters to capture the object's high-frequency details and places them in the target region, followed by a ControlNet-style Zhang et al. (2023) UNet encoder to refine and enhance object appearance. MV-Adapter Huang et al. (2024a) is an object-level customized multi-view image generation model. It is a plug-and-play adapter that employs a decoupled attention mechanism, learning inter-view consistency through a parallel architecture combining reference image cross-attention and multi-view attention.

# B ADDITIONAL EXPERIMENTAL DETAILS AND RESULTS

## B.1 PERSPECTIVE VIEW GENERATION FROM PANORAMA

We extract multiple perspective views based on predefined yaw ($\theta$) and pitch ($\phi$) angles. Each perspective view is generated using a fixed field of view (FOV) of $90°$ and an output resolution of $512 \times 512$.

We adopt two settings:

*5-view setting.* Five views are extracted at equal intervals along the horizontal plane with

$$\theta = [0°, \ 72°, \ 144°, \ 216°, \ 288°]$$
$$\phi = [0°, \ 0°, \ 0°, \ 0°, \ 0°].$$

*15-view setting.* A total of 15 views are extracted from three vertical levels (top, middle, bottom), each with 5 evenly spaced directions:

$$\theta = [0°, 72°, 144°, 216°, 288°, 0°, 72°, 144°, 216°, 288°, 0°, 72°, 144°, 216°, 288°]$$
$$\phi = [60°, 60°, 60°, 60°, 60°, 0°, 0°, 0°, 0°, 0°, -60°, -60°, -60°, -60°, -60°]$$

## B.2 DATA AUGMENTATION DETAILS

To enhance data diversity and improve the robustness of our model, we apply data augmentation to each reference image with a probability of 80%. The augmentation pipeline includes resizing, padding, random cropping, rotation, horizontal flipping, and mild geometric transformations, detailed as follows:

- **Resizing and padding**: Each input image is resized to fit within a target resolution while preserving its aspect ratio. The resized image is then padded with a white background to the exact target size.

- **Random half cropping**: With 20% probability (among the augmented samples), only half of the image is retained—either the left or right half, chosen randomly. The cropped region is aligned to one side and vertically centered on the padded canvas to simulate partial views and enhance object completion ability.

- **Random rotation**: If the image is not half cropped, it is randomly rotated within ±15° with a 70% probability, using white padding to fill any resulting empty regions.

- **Horizontal flip**: Also applied with 30% probability (if not half cropped), this operation mirrors the image horizontally to improve invariance to viewpoint direction.

- **Geometric transformations**: With 30% probability (again, if not half cropped), we apply a mild geometric transformation—either a perspective or affine transformation—with deformation limited to 5% of image dimensions.

## B.3 TRAINING AND INFERENCE DETAILS

**Reference images filter.** We specifically filter for objects belonging to the following categories: chair, bed, couch, potted plant, tv, car, toilet, dining table, and sink, retaining only those instances with detection confidence scores above 0.82. To ensure high-quality reference images, we apply additional size constraints: each region of interest (ROI) must have a longer dimension exceeding 100 pixels and a shorter dimension greater than 50 pixels. The selected ROIs then undergo background removal, resulting in clean, object-focused reference images optimized for customized generation tasks.

For both 5-view and 15-view models, we adopt stable-diffusion-2-1-base as the foundational generative model, with an image resolution of $512 \times 512$. Training is conducted using the AdamW optimizer with a learning rate of $1 \times 10^{-4}$. A DDPM scheduler is employed for noise scheduling during both training and inference. We adopt a two-stage training strategy: the model is first trained on the 5-view setting for 50 epochs using prompts where the category name is replaced by the corresponding image embedding; then, the category text is added and the model is trained for 1 additional epoch, as illustrated in Figure 9. The 15-view model is initialized from the 5-view model and further trained. The 5-view model was trained using four 24GB NVIDIA RTX 3090 GPUs, and the 15-view model was trained using four 48GB NVIDIA RTX 4090 GPUs.

In both training and inference phases, if the reference image category name does not appear in the prompt, the reference image feature is not embedded into the textual prompt, and the reference image is only introduced through the dual-UNet architecture by feeding it into each branch separately.

We provide the anonymized implementation of our method at the following github repository: `https://anonymous.4open.science/r/test-QM5E/`.

## B.4 ABLATION STUDY ON MULTI-MODAL PROMPT STRATEGY

To evaluate the effectiveness of our proposed multi-modal prompt strategy, we conduct ablation experiments under three different configurations:

Figure 9: Multi-modal prompts at different training stages.

- **Text-only:** The model is trained using only textual prompts, without incorporating any image features.
- **MM-single (single-stage multi-modal training):** The model is trained using textual prompts embedding image features during a single-stage training process.
- **MM-twostage (our two-stage multi-modal training):** Our adopted two-stage multi-modal training strategy, which first replaces the category name with the corresponding image embedding, then adds the category text and further trains the model.

The results of these different strategies are presented in Table 4.

Table 4: Comparison of different training methods in 5-view setting.

| Module | FID↓ | IS↑ | CS↑ | PSNR↑ |
|---|---|---|---|---|
| Text-only | **19.97** | 6.20 | **25.86** | 11.09 |
| MM-single | 20.83 | 6.38 | 25.76 | 11.08 |
| Ours(MM-twostage) | 20.91 | **6.70** | 25.79 | **11.40** |

We further validate the effectiveness of text-only and our MM-twostage on the more challenging 15-view setting. Results across different numbers of views are reported in Table 5.

Table 5: Comparison of different training methods in 15-view setting.

| Method | 15 views | | | 5 views | | | 8 views | | |
|---|---|---|---|---|---|---|---|---|---|
| | FID ↓ | IS ↑ | CS ↑ | FID ↓ | IS ↑ | CS ↑ | FID ↓ | IS ↑ | CS ↑ |
| Text-only | 21.91 | 6.79 | 27.39 | 26.86 | **7.36** | 28.00 | 25.42 | **6.68** | 26.74 |
| Ours(MM-twostage) | **20.93** | **6.92** | **27.51** | **25.65** | 6.76 | **28.01** | **23.55** | 6.16 | **26.80** |

Based on the above results, we adopt our MM-twostage training strategy in the final model.

B.5   ABLATION STUDY ON KEY SOURCE DESIGN IN HYBRID ATTENTION

Although our hybrid attention mechanism already outperforms the parallel structure (see main text), we further ablate the internal design of attention input sources. Specifically, we compare the following two configurations:

- **KV-from-Ref:** Queries are taken from the denoising branch, while keys and values are taken from the reference image features.
- **Ours:** Queries and keys are taken from the denoising branch, while values are taken from the reference image features.

The quantitative comparison is shown in Table 6. We observe that forcing both K and V to come from the reference image leads to noticeable performance degradation.

Table 6: Ablation study on key/value source design in hybrid attention.

| Method | FID↓ | IS↑ | CS↑ | PSNR↑ |
|---|---|---|---|---|
| KV-from-Ref | 21.24 | 6.41 | 25.73 | 10.76 |
| Ours | **20.91** | **6.70** | **25.79** | **11.40** |

### B.6 LOCATION-UNCONSTRAINED GENERATION

Our original model structure is designed to generate images from fixed viewpoints, with the primary intent of allowing users to flexibly place objects at specific positions within certain viewpoints while the model autonomously completes the rest of the scene. To accommodate application scenarios where predefined object locations are not desired, we also provide an additional version that does not require predefined object positions. Specifically, the model is modified by replacing the inputs of the reference branch with the ROIs (Regions of Interest) of the reference objects and removing the reference image inputs from the denoising branch. With this enhancement, users only need to provide the viewpoint where the reference image is to be inserted, and the model will adaptively adjust the object's position. Consequently, these enhancements address practical application issues and provide a more flexible solution for various scenarios.

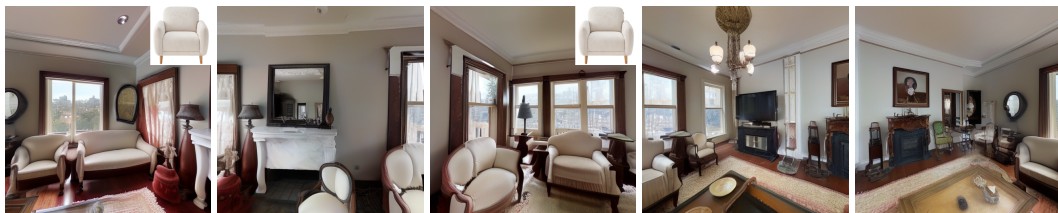

Figure 10: Visualization of location-unconstrained generation. The model determines the optimal placement for a given reference object.

### B.7 PROMPT FORMAT

Our recommended prompt format is to write one sentence describing each viewpoint, then concatenate all sentences together separated by commas. For example, in a 5-view setting (the same applies to 15-view), the individual viewpoint descriptions are:

- View 1: *A view of a large living room with a chandelier and a piano.*
- View 2: *There is a large room with a black dresser and a mirror.*
- View 3: *A close-up of a door leading to a hallway with a bench.*
- View 4: *There is a door that opens to a courtyard.*
- View 5: *There is a black dresser in a room with a mirror.*

These sentences are then combined into a single prompt:

*A view of a large living room with a chandelier and a piano, there is a large room with a black dresser and a mirror, a close-up of a door leading to a hallway with a bench, there is a door that opens to a courtyard, there is a black dresser in a room with a mirror.*

### B.8 ABLATION STUDY ON PANORAMA COORDINATE GUIDANCE

We also conducted an ablation experiment by removing the panorama coordinate guidance. As shown in Figure 11, the absence of this guidance leads to severe inconsistencies across different

viewpoints, significantly affecting spatial alignment. This validates the importance of panorama coordinate guidance in maintaining viewpoint consistency.

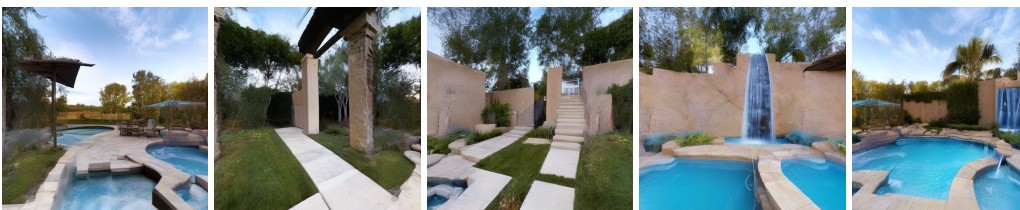

Prompt: a view of a pool with a patio and a patio furniture area, a view of a courtyard with a stone walkway and a stone patio, a close up of a house with a stone walkway and a stone wall, a view of a pool with a waterfall and a rock wall, a view of a pool with a waterfall in the middle of it

Figure 11: Ablation study on the effect of panorama coordinate guidance. Removing this guidance causes severe inconsistencies and spatial misalignment across viewpoints, demonstrating its crucial role in maintaining multi-view coherence.

## B.9    ADDITIONAL QUALITATIVE RESULTS

Figures 12-13 show additional qualitative results.

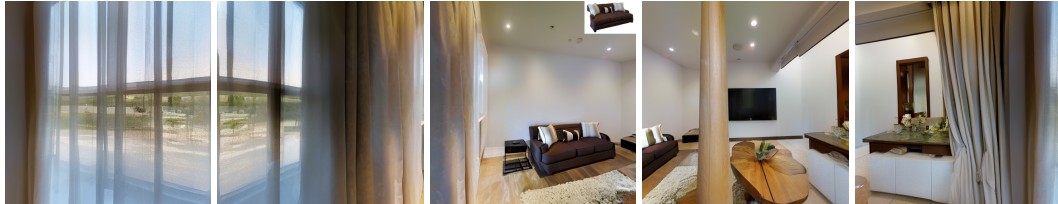

Prompt: there is a bird sitting on a wooden pole in the bathroom, a close up of a window with a curtain on it, arafed living room with a couch, chairs, coffee table and a large window, arafed view of a living room with a couch, coffee table, and television, blurry photograph of a blurry image of a wooden table

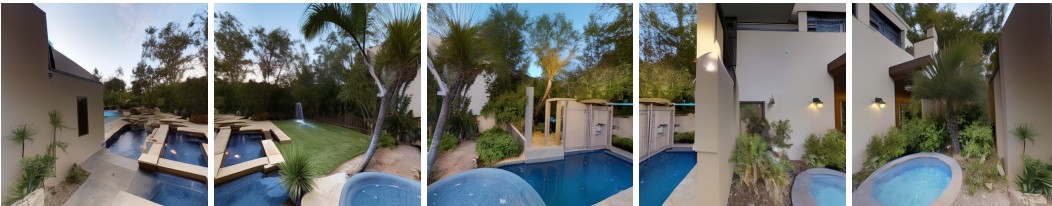

Prompt: there is a large yard with a stone walkway and a stone patio, a view of a patio with a fire pit and a pool, a view of a pool with a waterfall and a house in the background, a view of a pool with a waterfall and a rock wall, a view of a backyard with a pool and a fence

Figure 12: Additional qualitative results under the 5-view setting.

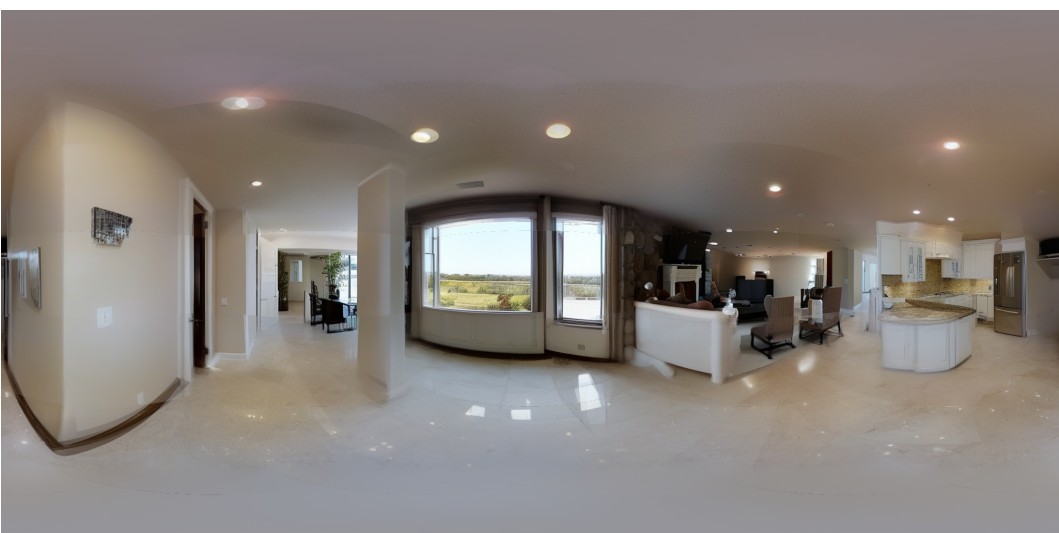

Prompt: there is a ceiling with a ceiling fan and a ceiling light, there is a ceiling with a clock on it in a room, there is a bathroom with a toilet and a window in it, a view of a ceiling with a light fixture and a window, there is a ceiling with a light that is on in a room, there is a room with a large wooden floor and a large window, a view of a living room with a couch and a table, a view of a kitchen with a refrigerator and a wooden floor, a view of a hallway with a sink and a mirror, a view of a hallway with a ceiling fan and a dining room, a view of a bathroom with a marble floor and a toilet, a view of a hallway with a marble floor and a white chair, there is a view of a bathroom with a marble floor, a view of a hallway with a marble floor and a wooden floor, a view of a bathroom with a marble floor and a wooden floor

Figure 13: Additional qualitative comparisons under the 15-view setting.

