# OpenReview forum: "Panorama Generation with Multiple Custom Background Objects"
_ICLR.cc/2026/Conference — ICLR 2026 Conference Withdrawn Submission_

### Official Review · Reviewer_yeAY · 2025-10-17

**Soundness:** 3
**Presentation:** 3
**Contribution:** 2
**Rating:** 4
**Confidence:** 5

**Summary:**

This paper addresses the problem of customized 360° panorama generation given multiple reference images and a text prompt. Unlike existing subject-driven customization or multi-view object reconstruction tasks, this work focuses on embedding multiple, weakly correlated reference objects into panoramic scenes while maintaining cross-view consistency and faithful object appearance.
To tackle this challenging setting, the authors propose a diffusion-based framework featuring:
- a dual-UNet architecture with decoupled feature injection, allowing selective reference conditioning only for views that include reference objects;
- a hybrid attention mechanism where queries/keys come from the denoising branch and values from the reference branch, balancing detailed reference injection with inter-view coherence;
- a two-stage multimodal prompt training strategy combining CLIP image embeddings with text guidance;
- data augmentation for viewpoint adaptation; and
- panorama coordinate guidance to maintain spatial consistency without explicit camera parameters.

Experiments on the Matterport3D dataset demonstrate that this method achieves superior or comparable performance to several state-of-the-art baselines, while being more efficient in inference. Ablation studies confirm the necessity of hybrid attention, data augmentation, and coordinate guidance for robust generation.

**Strengths:**

1. Novel task setting: The proposed task—customized panorama generation with multiple weakly correlated reference images—is indeed new. To the best of my knowledge, there are no prior works focusing on reference-based customization specifically within the panoramic image domain.
2. Architectural originality: The combination of dual-UNet and hybrid attention is a clear architectural innovation that directly targets the challenges of weakly correlated multi-reference conditioning and cross-view coherence.
3. Comprehensive design and analysis: The paper includes well-structured ablation studies and clear motivation for each proposed component (feature injection, coordinate guidance, and multimodal prompting), which strengthens the technical soundness of the approach.

**Weaknesses:**

1. Limited generation quality: The visual results are not convincing enough. Even in the authors’ own examples (e.g., Fig. 4), notable distortions appear, such as the vase on the table being heavily warped. These artifacts undermine the claim of achieving high-fidelity panoramic generation.
2. Insufficient experimental comparison:
  - The baseline selection is outdated. Key recent panorama or world-generation systems such as HunyuanWorld 1.0 (2025/07) and WorldGen (2025/04) are not compared, which weakens the empirical validity of the claimed superiority.
  - More critically, for the customization aspect of the task, no direct comparisons are made with existing customization approaches. For example, integrating IP-Adapter or similar modules with PanFusion or MVDiffusion would have provided a fairer and more competitive baseline. Without such comparisons, it is difficult to assess the real contribution of the proposed method.
3. Background leakage artifacts: In Fig. 5, clear background leakage appears, particularly around the leaf boundaries, producing visible white edges. This indicates limitations in spatial masking or blending. The authors could potentially mitigate this issue by applying an additional mask-based refinement or compositional filtering step.

**Questions:**

1. Please provide improved and more convincing visual results that better demonstrate the effectiveness and fidelity of your method.
2. Please add two supplemental experiments:
  1. comparisons to more recent panorama baselines or a brief explanation why they were omitted;
  2.  a customization baseline with PanFusion/MVDiffusion for a small-scale test—to clarify the gain from your proposed injection/attention design.
3. Some generated results show background leakage or object distortion. Can you try to solve it?

---

### Official Review · Reviewer_R5h5 · 2025-10-27

**Soundness:** 2
**Presentation:** 2
**Contribution:** 2
**Rating:** 2
**Confidence:** 3

**Summary:**

This paper proposes a model for a new task, customized panorama generation. Specifically, authors propose a multi-view diffusion model to generate multi-view images of the scene given text prompt. And the model can take additional object images and the inserted image as the "reference images" to generate multi-view images with the desired objects in it. Authors propose decoupled feature injection and hybrid attention architecture to incorporate with the reference image conditions. Extensive experiments show the superior performance of the proposed model comparing to other text-based panorama generation methods.

**Strengths:**

The motivation of the paper is clear and interesting: when generating panorama, only text prompt may not be enough in many cases, then reference object images can be a very effective way to enable more conditions from users.

Authors also propose effective architecture to fuse the different kinds of conditions, to achieve desired performance. They have done extensive experiments to compare with a series of baselines, and also ablate each proposed module.

The paper also provides detailed analysis on the intuition of each component, and the clarifications about many details in the paper.

**Weaknesses:**

The main weakness of the paper is that, although the proposed task seems interesting and meaningful, but the authors didn't show the proper experiments to validate the effectiveness of the proposed method, and the potential use of this new functionality. The authors also missed some important design details regarding the "customized panorama generation".

1. From the Figure 2. and also visual results in the experiment section, seems the reference image for one object, is always just "one image". Can two objects be placed in the same image? Can the same object be placed in two different images? If can't, what will be the case when desired object is located in overlapping region of two images.

2. Seems the task is simply trained by cropping out some objects from the training images, and use these images as "reference image control". In the trained model's results, seems the object is like to be "directly pasted inside the image", and while other contents in the image via text prompt, there exists severe texture mismatching problem between the object and other contents, e.g., Figure 5. Ours.

3. To actual show the effectiveness of the proposed model, I'd suggest authors show (qualitative) results of just simply replacing the reference image with novel instances, and keep everything else unchanged, to prove the model actually learn the desired task. This can be as simple as just replacing the reference image of a car with a new car.

4. This may be hard to quantitatively compare, but there exists another line of baselines, e.g. first use reference image to generate one view of the scene, and then use a scene-based multi-view diffusion model to generate multi-views. For example, ZeroNVS[1], I'd recommend the authors at least mention this type of baselines as a potential workflow.

[1] Sargent, Kyle, et al. "Zeronvs: Zero-shot 360-degree view synthesis from a single image." CVPR. 2024.

**Questions:**

Mostly from aforementioned weakness part. Additionally:

1. The data augmentation part is lack of clear introduction on what exactly part does this being applied to. Authors can show some visual examples of how to use augmentation in the training.

2. Some typos, e.g. in the beginning of the supplementary, "RALATED WORK" --> "RELATED WORK"

---

### Official Review · Reviewer_3n3Y · 2025-11-01

**Soundness:** 2
**Presentation:** 1
**Contribution:** 1
**Rating:** 2
**Confidence:** 3

**Summary:**

The paper presents a method for customized panorama generation: given a set of reference images and a textual prompt, it generates a 360° panorama as a sequence of views with cross-view correspondences, while aiming to place the reference objects naturally and faithfully at user-specified coordinates. To this end, the authors propose a dual-UNet framework in which a denoising branch is conditioned on features extracted by a reference-image branch via a hybrid-attention mechanism. They compare against several panorama-generation baselines under 5- and 15-view settings and, in most cases, report superior quantitative performance.

**Strengths:**

* The paper tackles an interesting problem: generating multi-view panoramas that incorporate multiple reference objects at specified locations.

* Quantitatively, the method generally outperforms several panorama baselines on standard image-quality metrics across 5, 8, and 15-view configurations.

* Qualitative samples include several cases with coherent, visually plausible placement of the reference objects.

**Weaknesses:**

* Novelty: The central architectural idea - “hybrid attention” that conditions a denoising U-Net on features from a separate reference-image branch - appears closely related to MV-Adapter–style attention. The novelty over this line of work seems incremental, and the ablation indicates only marginal quantitative gains from the proposed variant.

* Presentation: The paper would benefit from being more self-contained. Key implementation details are missing or hard to locate: the exact diffusion backbone(s), sampling/training schemes, and how these are adapted to multi-view are not specified in the main text. A formal description of the training objective and inference procedure for the diffusion model is absent.  The inputs/outputs and data flow of the attention modules in Eqs. 2–4 are difficult to parse; a clearer diagram or pseudocode would help. The data-augmentation strategy, which seems important to performance, is under-explained in the main paper. A concise section summarizing limitations and future work is currently missing.

* Baselines:  A straightforward baseline—adapting a pretrained multi-view diffusion model to this task via personalization (e.g., Textual Inversion, DreamBooth) or light finetuning—is not included. Without such comparisons (or close proxies), it is hard to gauge the practical significance of the reported improvements.

* Qualitative Evaluation: The paper presents relatively few qualitative results, which makes it difficult to assess the strengths and failure modes of the approach. Some shown examples exhibit noticeable artifacts (e.g., Fig. 5: visible white background behind the plant, cross-view texture inconsistencies on pillows, and horizontal line artifacts). In these cases, some baselines appear visually superior. For fairer assessment, it would be preferable to show full panorama renderings for both the method and baselines (e.g., Figs. 4 and 6), rather than only pre-stitching views.

**Questions:**

* Please elaborate on how the proposed architecture differs from prior multi-view conditioning approaches (e.g., MV-Adapter–style attention). What components or design choices are new, and why are they necessary for this task?

* Could the authors provide a clearer description of the hybrid attention module, compared to Eq. 2-4?
Please specify inputs/outputs, tensor shapes, where it is inserted in the U-Net(s), and how gradients flow between branches.

* Why is the method better suited to this task than adapting a pretrained multi-view diffusion model via personalization (e.g., Textual Inversion, DreamBooth) or light finetuning?
If feasible, please include or discuss results for these baselines (or close proxies), and clarify expected failure modes where your approach has an advantage.

* Can the authors provide more full-panorama comparisons against baselines ?

---

### Note · Authors · 2025-11-12

I have read and agree with the venue's withdrawal policy on behalf of myself and my co-authors.